# A CMOS Image Readout Circuit with On-Chip Defective Pixel Detection and Correction

**DOI:** 10.3390/s23020934

**Published:** 2023-01-13

**Authors:** Bárbaro M. López-Portilla, Wladimir Valenzuela, Payman Zarkesh-Ha, Miguel Figueroa

**Affiliations:** 1Electrical Engineering Department, University of Concepción, Edmundo Larenas 219, Concepción 4070386, Chile; 2Department of Electrical and Computer Engineering (ECE), University of New Mexico, Albuquerque, NM 87131-1070, USA

**Keywords:** defective pixel, detection and correction algorithms, image sensor, integrated circuit, intelligent readout circuit

## Abstract

Images produced by CMOS sensors may contain defective pixels due to noise, manufacturing errors, or device malfunction, which must be detected and corrected at early processing stages in order to produce images that are useful to human users and image-processing or machine-vision algorithms. This paper proposes a defective pixel detection and correction algorithm and its implementation using CMOS analog circuits, which are integrated with the image sensor at the pixel and column levels. During photocurrent integration, the circuit detects defective values in parallel at each pixel using simple arithmetic operations within a neighborhood. At the image-column level, the circuit replaces the defective pixels with the median value of their neighborhood. To validate our approach, we designed a 128×128-pixel imager in a 0.35μm CMOS process, which integrates our defective-pixel detection/correction circuits and processes images at 694 frames per second, according to post-layout simulations. Operating at that frame rate, our proposed algorithm and its CMOS implementation produce better results than current state-of-the-art algorithms: it achieves a Peak Signal to Noise Ratio (PSNR) and Image Enhancement Factor (IEF) of 45 dB and 198.4, respectively, in images with 0.5% random defective pixels, and a PSNR of 44.4 dB and IEF of 194.2, respectively, in images with 1.0% random defective pixels.

## 1. Introduction

Image sensors are devices that convert incident light into electrical signals using photosensitive elements [1]. These devices are at the heart of every imaging system, and their core operation consists of two steps: first, the image sensor converts the incident photons into electron–hole pairs, and then, it transforms the resulting current into a voltage [2]. The voltage at each photosensitive element is proportional to the number of photons received during the acquisition time [3].

Currently, Charge Coupled Device (CCD) and Complementary Metal Oxide Semiconductor (CMOS) image sensors are the preferred technologies for digital imaging devices [4,5]. Although CCDs often deliver better image quality and low-light robustness, CMOS Image Sensors (CIS) are preferred on mobile applications due to their higher dynamic range and smaller current and voltage requirements [5]. Furthermore, recent advances in deep submicron CMOS have brought significant advantages for CIS, such as compatibility with standard CMOS technology, the possibility of integrating sensors with analog and digital circuits for image processing at the pixel-level, random access to image data, low power consumption, and high-speed image acquisition [4,5,6,7,8]. However, they frequently contain defective pixels as a result of fabrication errors or device malfunction due to sensor wear and tear [9,10]. This problem is exacerbated by the increasing demand for high-resolution and high-density imagers [11,12,13]. Therefore, it is essential to detect and correct these defects at early processing stages in order to produce images that are useful to human users as well as to image processing and machine vision algorithms [13,14,15].

Defective pixels are grouped into three classes: hot, dead, and stuck [16]. The first two are used for grayscale imagers, while the third is used in multi-color sensors where one or more sub-pixels (e.g., red, blue, or green) are hot or dead. Dead pixels generate a zero or near-zero value regardless of the incident light intensity. In contrast, a hot pixel responds to all illumination levels with a high output value.

Many defective pixel detection and correction algorithms have been reported in the literature [12,13,17,18,19]. Most algorithms that operate online using data from a single image detect defective pixels by computing statistical variations within a neighborhood centered around each pixel. Although these algorithms frequently run on programmable architectures such as traditional computers and embedded processors, custom hardware architectures can provide higher performance with lower resources and power consumption, which are required for applications such as assisted-driving automobiles [20,21,22], surveillance systems [23,24,25], and biometric recognition [26,27,28], among many others. Numerous researchers have proposed custom hardware devices to perform image processing in real time, which reduces power consumption and increases hardware integration. Examples of such devices are Smart Image Sensors (SIS) [29], which integrate image capture and processing on the same die. The image processing hardware can be integrated at the pixel, column, or chip level [3], increasing the parallelism available to the algorithm and throughput of the implementation while reducing latency, memory requirements, and power consumption [30,31].

In this paper, we propose a novel CMOS SIS architecture for on-chip defective pixel detection and correction using pixel- and column-level processing with analog circuits. The SIS architecture implements a modified version of the algorithm proposed by Tchendjou et al. [12], which simplifies the computation to make it more suitable for custom hardware implementation. The SIS circuits detect all defective pixels in the image in parallel by performing arithmetical operations at the pixel level during photocurrent integration. Column-level circuits correct defective pixels by computing the median value of a neighborhood. A physical design of the SIS using a 0.35μm CMOS process can acquire and process 128×128-pixel images at 694 frames per second (fps). Using 500 images from the Linnaeus 5 dataset [32] with randomly added defective pixels, the proposed algorithm and its analog-hardware implementation achieve better performance than the best single-frame detection and correction algorithms available in the literature [12,13,17,18,19].

The rest of this paper is organized as follows: Section 2 discusses related works; Section 3 presents our defective pixel detection and correction algorithms; Section 4 describes the design of the SIS; Section 5 evaluates the performance of the proposed algorithm and the SIS; and Section 6 presents our conclusions.

## 2. Related Works

The defective pixel rate is one of the most significant characteristics used to evaluate the quality of an image sensor at the end of its manufacturing process and during its useful life [13,33,34]. The occurrence of defective pixels results in degradation of the overall visual quality of the captured image, which can have a critical effect on its subsequent interpretation, especially if additional processing is performed on the image. For example, in astronomy, it is crucial to discriminate between a hot pixel and a pixel that represents a bright star [10], while in mammographic imaging defective pixels can be confused with microcalcifications [35]. Therefore, defective pixel detection and correction algorithms are crucial for correctly interpreting the information in the image. In this work, we focus on online algorithms, which detect defective pixels in the image sensor from one or more video frames acquired during normal operation in real time [9,13,36]. In particular, we focus on single-frame defective pixel detection algorithms, which are faster and require less memory than multi-frame algorithms.

Chan [17] proposed a dead pixel detection method which labels a pixel as defective when its value lies outside of an interval defined as [a−r,a+r], where *a* is the average of its eight neighbors except for the second-to-largest and second-to-smallest values and *r* is the range between these two values. Cho et al. [18] detected defective pixels based on whether the difference between its value and the average between two of its neighbors is greater than the mean of the pixel and its four neighbors or the maximum pixel value minus this mean. These algorithms use only basic operations such as absolute difference, comparison, spatial neighborhood averages, and rank-ordering. Although they can be implemented efficiently, these algorithms produce high false positive rates, increasing the number of pixels corrections and degrading the quality of the resulting image [12]. El-Yamany [37] showed a robust method of identifying singlets and couplets of defective pixels that uses comparison, averaging, minimum, and maximum operations. They used two tunable parameters that control the detection sensitivity and specificity. Tchendjou et al. [12] proposed an algorithm that flags defective pixels when the difference between the actual and estimated value is greater than a threshold (or the maximum pixel value minus this threshold). The threshold is computed as the mean between the pixel value and a simple average of its eight neighbors. The estimated value is computed as a weighted (or in the simplified version, unweighted) average of the adjacent pixels. They achieved better sensibility, specificity, positive predictive value, and phi-coefficient than [17,18]. The authors of [38] proposed an optimization algorithm to detect and correct defective pixels in an image using two defective pixel detection methods; the first uses the distance between the pixel and its neighbors [12], while the second uses the dispersion of the pixel values within a neighborhood window [39]. Yongji et al. [19] used a method similar to Chan [17] for defective pixel detection, except that they used a 5×5-pixel window to compute the average and excluded the two largest and the two smallest pixels in this window.

The algorithms described above can achieve good accuracy; however, they must be programmed on high-performance processors in order to achieve short processing times. While the subsequent cost in die area and power consumption of these solutions may not be a limitation in larger applications, they are typically unsuitable for portable and mobile devices. Consequently, many researchers have proposed special-purpose processing systems based on custom hardware architectures in order to focus on speed, portability, and low power consumption [13,36,39,40]. In fully-digital solutions, FPGAs (Field Programmable Gate Array) are a widely used platform to implement these architectures thanks to their large fine-grained parallelism and relatively low cost and power consumption compared to traditional programmable processors. Kumar et al. [40] implemented real-time non-uniformity correction and defective pixel detection on a Xilinx XC2S1500 FPGA using a thresholding criterion based on the mean ± 3 times the standard deviation of the pixel values within a neighborhood. In [36], the authors implemented a combination of offline and online defective pixel detection and correction on a Lattice Semiconductors ECP2 FPGA. In each frame, they computed the difference between the current pixel and the average of its neighbors and then compared it to a threshold in order to select candidate defective pixels. If the same pixel was detected in consecutive frames, that pixel was considered to be defective. In [39], the authors used the median, standard deviation, range, interquartile range, and other statistical dispersion parameters of pixel blocks to detect and correct defective pixels in an image. Running their algorithm on a Microblaze microcontroller embedded in a Xilinx VC707 FPGA, the quality of their results was only slightly lower than those in [12] while achieving a 3.5 times faster computation time. Tchendjou et al. [13] presented a heterogeneous architecture on a Xilinx ZC702 device implementing three detection algorithms: the first was the algorithm presented in [12]; the second was similar to the first, except that it computed the estimated pixel value as the median of the neighborhood window; the third, described in [39], was faster and able to achieve slightly better results than the first two, though it used more hardware resources. Moreover, the last algorithm required five parameters to be determined empirically based on the images in the dataset.

Despite the significant advantages of the architectures described above, these custom digital processors operate outside the CIS, accessing the pixel values in sequential fashion after analog-to-digital conversion. This limits the intrinsic data parallelism of the algorithm, reducing the throughput and increasing the latency of the implementation. Moreover, sequential access to the pixel values requires the use of line buffers in the digital processor to implement window-based operations. As an alternative, all or part of the operations of the algorithm can be performed in an SIS at the pixel or column level, usually via analog circuits integrated into the image sensor [41,42,43]. This approach greatly increases the parallelism of the implementation, though at the cost of increasing the pixel size or decreasing the area available for photodetectors. Ginhac et al. [44] reported a dynamically configurable parallel processor array integrated into a CMOS image sensor. A single-instruction multiple-data architecture enabled programmable mask-based image processing computation on each pixel. Garcia-Lamont [45] described an analog SIS to compute the Prewitt edge detection algorithm using a simple arithmetic analog circuit integrated with each photodetector in the imaging array. Suárez et al. [46] presented an SIS for Gaussian pyramid extraction with per-pixel analog processing circuits and local analog memories. Gottardi et al. [47] proposed a vision sensor that computes a four-bit Local Binary Pattern (LBP) for each pixel during integration time. They demonstrated the use of these LBPs to perform software-based image description and retrieval.

To reduce the impact of pixel-level processing on the pixel area and photodetector utilization, a number of works have been proposed in the literature implementing computation at the column level [48,49,50,51,52]. Young et al. [48] presented an SIS that used column-parallel readout analog circuits to compute histograms of oriented gradients for object detection. Jin et al. [49] proposed an edge detection SIS that used column-level analog circuits and digital static memory to compare pixels in adjacent columns. In [50], the authors presented a vision sensor that computed frame differences and background subtraction for motion sensing and object segmentation. Hsu et al. [51] presented an SIS for feature extraction that computed the convolution between the image and a 3×3 kernel, with the programmable weights represented as analog currents and the pixels as pulse-width modulated voltages. In our own previous work [52,53], we combined pixel- and column-level analog computation during photocurrent integration using a custom digital processor. In [52], the analog circuits were used to compute a simplified version of LBP for use by the digital coprocessor in performing facial recognition on images. In [53], we used pixel-level analog memory to compute frame differences and motion estimation, with the digital processor using a connected components algorithm to detect objects in motion.

The research described above shows the importance of detecting and correcting defective pixels before displaying or processing an image. While most hardware implementations of these algorithms have used digital programmable logic devices, research shows that implementing the computation at the pixel or column level can greatly improve the parallelism of the implementation. Moreover, performing computation in the analog domain during photocurrent integration can reduce the impact of device mismatch and nonlinearity on the accuracy of the results and lower the area overhead of the computation circuits.

## 3. Methods

The circuits in our SIS implement a detection and correction algorithms for defective pixels. We detect defective pixels by comparing each pixel value to its four neighbors (north, south, east, and west). The correction algorithm simply replaces the defective pixel value with the median of the same neighbors. In the rest of this section, we describe our proposed pixel detection algorithm.

Our proposed algorithm for detecting defective pixels modifies the algorithm presented in [12] to make it more suitable for implementation on analog hardware. The original algorithm operates within a single-image frame, first defining a 3×3-pixel window centered on the pixel under test, denoted as $P_{0}$, as shown in Figure 1. If $P_{0}$ has neither the smallest nor the largest value in the window, the algorithm assumes that the pixel is not defective and continues with the next pixel in the image. Otherwise, $P_{0}$ is marked as possibly defective and the algorithm computes an estimate for its value as the average of its eight neighbors, as described in Equation (Equation 1):(1)$$P_{est}=\frac{1}{8}\sum_{i=1}^{8}P_{i},$$
where $P_{est}$ is the estimated value for $P_{0}$ and $P_{1}–P_{8}$ are its neighboring pixels in the 3×3-pixel window. Then, the algorithm computes the average and difference between $P_{0}$ and $P_{est}$, as described in Equations (Equation 2) and (Equation 3): (2)$$\begin{array}{ccc} P_{avg} & = & \frac{P_{est}+P_{0}}{2} \end{array}$$(3)$$\begin{array}{ccc} P_{diff} & = & \left|P_{est}-P_{0}\right|. \end{array}$$

The algorithm marks $P_{0}$ as a defective pixel when $P_{diff}$ is greater than the absolute difference between $P_{avg}$ and the minimum (0) or maximum ($P_{max}$) pixel value, as described in Equations (Equation 4) and (Equation 5): (4)$$\begin{array}{c} P_{diff}>P_{avg} \end{array}$$(5)$$\begin{array}{c} P_{diff}>P_{max}-P_{avg}, \end{array}$$
where $P_{max}$ is the maximum possible value of any pixel in the image.

Assuming that $P_{0}<P_{est}$ and substituting Equations (Equation 2) and (Equation 3) into Equations (Equation 4) and (Equation 5), the algorithm marks $P_{0}$ as a dead pixel when one of the conditions described in Equations (Equation 6) and (Equation 7) are met: (6)$$\begin{array}{c} P_{0}<\frac{1}{3}P_{est} \end{array}$$(7)$$\begin{array}{c} P_{0}<3\left(P_{est}-\frac{2}{3}P_{max}\right). \end{array}$$

Likewise, when $P_{0}\geq P_{est}$, substituting Equations (Equation 2) and (Equation 3) into Equations (Equation 4) and (Equation 5), the algorithm marks $P_{0}$ as a hot pixel when it meets one of the conditions in Equations (Equation 8) and (Equation 9):(8)$$P_{0}>3P_{est}$$
(9)$$P_{0}>\frac{1}{3}\left(P_{est}+2P_{max}\right).$$

In order to simplify the hardware implementation of the algorithm, we propose the following changes to the original algorithm:1.We use only four pixels in the 3×3-pixel neighborhood to compute the estimated value of $P_{0}$, as follows:
(10)$$P_{est}=\left(P_{2}+P_{5}+P_{7}+P_{4}\right)/4.$$2.Instead of using computation to determine whether $P_{0}$ is the pixel with the maximum or minimum value in the neighborhood, we mark $P_{0}$ as possibly dead when $P_{0}<P_{L}$ and possibly hot when $P_{0}>P_{H}$, with $P_{L}=0.05P_{max}$ and $P_{H}=0.95P_{max}$ [18,54,55].3.As shown in Figure 2a, when $P_{est}<0.05P_{max}$ and Equation (Equation 6) is true, then Equation (Equation 7) is always true. Therefore, we only use Equation (Equation 6) to detect dead pixels.4.Likewise, Figure 2b shows that when $P_{est}>0.95P_{max}$ and Equation (Equation 9) is true, Equation (Equation 8) is always true. Therefore, we only use Equation (Equation 9) to detect hot pixels.5.To compensate for any offset in the analog-circuit implementation of Equations (Equation 6) and (Equation 9), we introduce two adjustable thresholds, $P_{TL}$ and $P_{TH}$. We determine the value of these offsets in Section 5. Thus, we rewrite Equations (Equation 6) and (Equation 9) as Equations (Equation 11) and (Equation 12) for dead and hot pixels, respectively:(11)$$P_{0}-\frac{1}{3}P_{est}<P_{TL}$$
(12)$$P_{0}-\frac{1}{3}P_{est}-\frac{2}{3}P_{max}>P_{TH}.$$

In summary, Algorithm 1 shows the our proposed defective pixel detection algorithm. The next section describes the analog CMOS implementation of our detection and correction algorithm.   
**Algorithm 1:** Proposed detection method.
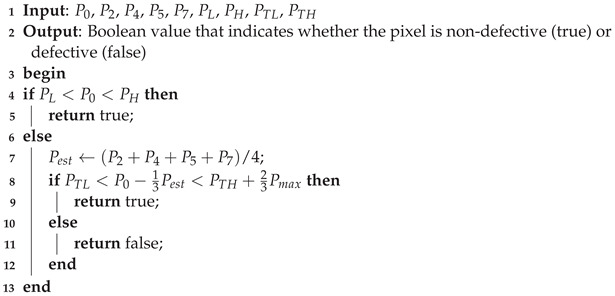


## 4. Defective Pixel Detection and Correction Circuits

In the proposed readout circuit, the defective pixel detection stage of the algorithm is implemented in the analog domain at the pixel level. The defective pixel correction stage operates at the column level, again in the analog domain. This allows us to reduce the number of transistors used to implement the algorithm, thereby increasing the fill factor of the imager (i.e., the ratio between the die area occupied by the photodetector and the area of the entire pixel circuit) and reducing its power consumption.

### 4.1. Defective Pixel Detection Circuit

Each pixel in the imager is composed of a photodetector and a defective pixel detection circuit. This circuit receives as input the current from its own photodetector PD0 and its four neighbors, PD2, PD5, PD7, and PD4, as shown in Figure 3, and produces a single-bit output that signals whether the pixel is defective or not. The circuit is based on a Capacitive Transimpedance Amplifier (CTIA), which integrates a photodetector current and produces a voltage that represents the intensity of the incident light at the pixel during the integration time. Compared to other readout circuits, the CTIA has a larger dynamic range and higher linearity due to the high gain of its Operational Transconductance Amplifier (OTA) [56,57]. We used a conventional two-stage OTA [58,59] for the CTIA and the two comparators.

The circuit shown in Figure 3 first compares the value of the current pixel $P_{0}$ to the limits $P_{L}$ and $P_{H}$, as required by Line Four of Algorithm 1, then evaluates Equations (Equation 11) and (Equation 12). The circuit output $V_{p}$ is a logic 0 when $P_{0}$ is a defective pixel.

During the integration time *T*, the circuit operates in two phases. Figure 4a shows the activation of the switches SWP0, SWP2, SWP5, SWP7, and SWP4 that select the input to the defective pixel detection circuit of Figure 3 during the integration time. As shown in the figure, Phase 1 operates during $T_{1}=3T/4$ and Phase 2 during $T_{2}=T/4$. Figure 4b shows the equivalent circuit of the defective pixel detector during Phase 1. Switch SWP0 in the Input Select block of Figure 3 is closed, and the other input select switches are open, providing the output current of photodetector PD0 as the input to the Custom CTIA block. In this block, switches SWC1 and SWC4 are closed; thus, the CTIA integrates the current generated by the photodetector PD0 in the positive direction. At the end of Phase 1, the output voltage of the CTIA is provided by Equation (Equation 13):(13)$$V_{C}=\frac{T_{1}}{C_{int1}}I_{0}+V_{REF},$$
where $C_{int1}$ is the integration capacitor value, $I_{0}$ is the PD0 photodetector current, and $V_{C}$ is the output voltage at the end of Phase 1. Choosing $V_{REF}=V_{dd}/2$ guarantees that the OTA output signal is symmetrical and has a maximum dynamic range. In the Comparators block, switches SWH and SWL are closed; thus, after $T_{1}$, the OTA output $V_{C}$ is compared against $V_{L}$ and $V_{H}$ using comparators COMP1 and COMP2, storing a logic 1 in flip-flop *FF1* if either $V_{C}<V_{L}$ or $V_{C}>V_{H}$ (Line Four of Algorithm 1). At the end of Phase 1, $V_{C}$ is stored in the sample-and-hold circuit shown at the bottom right corner of Figure 3 for later use by the defective pixel correction circuit.

Figure 4c shows the equivalent circuit during Phase 2. At the start of this phase, the voltage across the integration capacitor $V_{C}$ represents $P_{0}$. At this point, switches SWC1 and SWC4 are opened and switches SWC2 and SWC3 are closed, inverting the polarity of the output. The circuit divides Phase 2 into four equal time intervals with duration $T_{2}/4$. As shown in Figure 4a, within each time interval only one of switches SWP2, SWP5, SWP7, or SWP4 is closed at a time; thus, the CTIA successively integrates the current from photodetectors PD2, PD5, PD7, or PD4 in the negative direction. At the end of Phase 2, the output voltage $V_{C}$ is provided by Equation (Equation 14):(14)$$V_{C}=\frac{T_{2}}{4C_{int1}}\left(I_{2}+I_{5}+I_{7}+I_{4}\right)-\frac{T_{1}}{C_{int1}}I_{0}+V_{REF}$$
where $I_{2}$, $I_{5}$, $I_{7}$, and $I_{4}$ are the output currents of the photodetectors PD2, PD5, PD7, and PD4, respectively.

At this point, voltage $V_{C}$ represents $\frac{1}{3}P_{est}-P_{0}$. Therefore, instead of evaluating Equations (Equation 11) and (Equation 12), the Comparators block evaluates
(15)$$\begin{array}{ccc} \frac{1}{3}P_{est}-P_{0} & > & -P_{TL} \end{array}$$
(16)$$\begin{array}{ccc} \frac{1}{3}P_{est}-P_{0} & < & -\left(P_{TH}+\frac{2}{3}P_{max}\right), \end{array}$$
by closing switches SWTH and SWTL and opening SWH and SWL. In Figure 4c, the voltages $V_{TL}$ and $V_{TH}$ represent $-P_{TL}$ and $-(P_{TH}+\frac{2}{3}P_{max})$, respectively. The output of comparators COMP1 and COMP2 is a logic 0 when the corresponding conditions in Equations (Equation 15) and (Equation 16) are met, producing a logic 1 at the output of gate NAND1.

Finally, the NAND2 gate in Figure 3 produces the output $V_{P}$ of the defective pixel detection circuit, such that $V_{P}=0$ (which signals a defective pixel) when $P_{0}<P_{L}$ or $P_{0}>P_{H}$ and one of the conditions represented by Equations (Equation 15) and (Equation 16) (or equivalently, Equations (Equation 11) and (Equation 12)) are met.

### 4.2. Defective Pixel Correction Circuit

Figure 5 shows the block diagram of the defective pixel correction circuit. The inputs V2, V5, V7, and V4 are the output voltages $V_{c}$ of the configurable CTIA stage (see Figure 3) of the four neighboring pixels. The circuit is a median filter that computes the median value of its four inputs using a two-stage analog sorting network, followed by an output stage that computes the average value of the two central inputs $V_{A}$ and $V_{B}$ after sorting:(17)$$V_{out}=\frac{V_{A}+V_{B}}{2},$$
where
(18)$$\begin{array}{ccc} V_{A} & = & \min\left[\max\left(V2,V5\right),\max\left(V7,V4\right)\right] \end{array}$$
(19)$$\begin{array}{ccc} V_{B} & = & \max\left[\min\left(V2,V5\right),\min\left(V7,V4\right)\right]. \end{array}$$

The buffers are implemented using a single-stage OTA [58,59], and the sorting network is composed of circuits that compute the minimum and maximum value of their two inputs. Figure 6a and Figure 6b show the minimum and maximum circuits, respectively.

Figure 6 shows the circuits that compute the minimum and maximum operations used in Equations (Equation 18) and (Equation 19), which were proposed by Soleimani [60]. Both circuits consist of a differential amplifier stage composed of $M_{I1}$, $M_{I2}$, $M_{out}$, $M_{C1}$, $M_{C2}$, and $M_{Cout}$, along with a common-source stage with an active load, composed of $M_{P1}$, $M_{P2}$ and $M_{2}$. Because the operation of both circuits is similar, we explain only the operation of the minimum-value circuit shown in Figure 6a. When $V_{in1}<V_{in2}$, $M_{I2}$ is almost off and the drain currents of $M_{I1}$ and $M_{out}$ are equal to $I_{B}$ because of the current mirror formed by $M_{C1}$ and $M_{Cout}$. Due to the state of $M_{I2}$, the gate voltage of $M_{P2}$ is very small, and this transistor is almost turned off as well. Therefore, all of current $I_{B}$ from $M_{2}$ passes through $M_{P1}$. Because the current passing through $M_{P1}$ and $M_{Cout}$ has the same value $I_{B}$, the gate voltages $V_{GS}$ of both transistors are the same. Finally, the source and drain voltages and drain current of $M_{I1}$ and $M_{out}$ are equal, which results in $V_{out}=V_{in1}$. The same reasoning applies when $V_{in2}<V_{in1}$, in which case $V_{out}=V_{in2}$. The operation of the circuit in Figure 6b follows the same principle, except with computation of the maximum value between the two inputs $V_{in1}$ and $V_{in2}$.

### 4.3. General Readout Circuit

Figure 7 shows the 128×128 smart pixel array. Each smart pixel, denoted as P&DC, includes a photodetector and a defective pixel detection circuit. Each column includes a defective pixel correction circuit (CC) and a 2:1 analog multiplexer. An external digital device generates the timing and control signals to the array and the 128:1 analog multiplexer that generates the output.

Each P&DC circuit has two outputs; the first is the output value $V_{C}$ of the defective-pixel detection circuit at the end of Phase 1, and the second is the output value $V_{P}$ that indicates whether the pixel is defective or not. Each column reads the $V_{C}$ output of three smart pixels, which are the center $P_{0}$, top $P_{2}$, and bottom $P_{7}$ pixels of the 3×3-pixel window in Figure 1, as well as the $V_{P}$ output of the central pixel $P_{0}$. The pixel values $V_{C}$ are stored in a sample and hold circuit at the end of Phase 1. The defective pixel correction circuit uses the $P_{2}$ and $P_{7}$ outputs along with the $P_{4}$ and $P_{5}$ outputs of the neighboring columns to compute the corrected pixel value $V_{out}$. The 2:1 multiplexer selects between $P_{0}$ and the corrected value $V_{out}$ depending on the logic value of $V_{P}$, which indicates whether or not $P_{0}$ is defective. The 128:1 analog multiplexer serially outputs all the pixel voltage values within each row.

## 5. Results

### 5.1. Physical Layout

Figure 8 shows the physical layout of the defective pixel detection circuit on a 0.35μm mixed-signal CMOS processor with three metal layers, two polysilicon layers, and a 3.3V supply voltage.

We used a double-poly capacitor with a capacitance per unit area of 950 aF/μm${}^{2}$. Assuming an integration time *T* = 16 μs (Section 4.1) and a maximum photodetector current of 14 nA, the pixel requires a 100 fF integration capacitor of 13.1μm by 8.0μm. The area of the complete defective pixel detection circuit is 42μm by 63μm. Considering a 64μm by 64μm pixel [61], the circuit achieves a fill factor of 35.4%. In comparison, the fill factor without these additional stages is 73.4%. It is worth noting that only two wires are required between neighboring cells, each transporting photocurrent in a different direction. This is because, as described in Section 4.1, the defective pixel detection circuit implements Equations (Equation 10)–(Equation 12) using time multiplexing, reading the photocurrent of only one pixel at a time. Moreover, because at most only one of the two wires operates in low impedance at the same time, the effects of crosstalk are diminished.

Although the results presented in this section use the 0.35 μm physical design described above, we additionally evaluated the scaleability of our circuits by porting the 0.35 μm physical design to a standard 0.18μm CMOS processor frequently used in the literature [62,63,64,65]. The 0.18 μm processor uses a 1.8V supply voltage and features metal–insulator–metal capacitors with a capacitance per unit area of 2 fF/μm${}^{2}$. The total area of the defective pixel detection circuit in the 0.18 μm process is 30.1μm by 45.2μm, which achieves a fill factor of 66.7% in the same 64μm by 64μm pixel. Without the comparators and output logic stages, the fill factor is 86.3%.

Figure 9 shows the layout of the defective pixel correction circuit, which has an area of 42μm by 35μm in the 0.35μm processoror and 21.2μm by 17.8μm in the 0.18μm standard CMOS processor.

### 5.2. Algorithm Parameters

To select the values of $P_{TL}$ and $P_{TH}$ in Algorithm 1, we used 250 images with different levels of brightness, contrast, and dynamic range randomly selected from the Linnaeus 5 dataset [32]. To determine the value for $P_{TL}$, we randomly added 0.5% dead pixels to the image with values between 0 and $0.05P_{max}$. Similarly, we added 0.5% hot pixels (between $0.95P_{max}$ and $P_{max}$) to determine the value of $P_{TH}$.

We simulated the performance of the algorithm as a function of the values of $P_{TH}$ and $P_{TL}$ in the interval $\left[-0.025P_{max},0.025P_{max}\right]$. We evaluated the performance of the algorithm using four metrics: Sensitivity (Se), Specificity (Sp), precision or Positive Predictive Value (PPV), and Phi coefficient (ϕ), defined as follows: (20)$$\begin{array}{ccc} Se & = & \frac{TP}{TP+FN} \end{array}$$(21)$$\begin{array}{ccc} Sp & = & \frac{TN}{TN+FP} \end{array}$$(22)$$\begin{array}{ccc} PPV & = & \frac{TP}{TP+FP} \end{array}$$(23)$$\begin{array}{ccc} \phi & = & \frac{TP\cdot TN-FP\cdot FN}{\sqrt[{}]{\left(TP+FP)\cdot (TP+FN)\cdot (TN+FP)\cdot (TN+FN\right)}}, \end{array}$$
where TP is the number of true positive defective pixels in the image detected by the algorithm, TN is the number of true negatives, FP is the number of false positives, and FN is the number of false negatives; Sensitivity quantifies the fraction of pixels marked as defective by the algorithm that are truly defective; Specificity expresses the fraction of good pixels correctly identified as such by the algorithm; PPV is the precision of the algorithm, that is, the fraction of truly defective pixels that were detected; and the ϕ coefficient is a special case of the Pearson correlation coefficient that measures the difference between the ratios of correctly and incorrectly classified pixels.

Figure 10a and Figure 10b respectively show the four metrics averaged over the 250 images for a range of $P_{TL}$ and $P_{TH}$ values. Figure 10a shows the performance of the algorithm when detecting dead pixels, and Figure 10b depicts the performance metrics when detecting hot pixels. These graphs show that increasing the value of $P_{TL}$ and $P_{TH}$ improves the PPV and ϕ metrics and degrades Se, while the Specificity remains approximately constant at Sp=0.99. Based on these results, we chose $P_{TL}=0.012P_{max}$ and $P_{TH}=0.015P_{max}$, which resulted in a sensitivity of Se=0.95. With these values, we achieve ϕ=0.83 and PPV=0.75 when detecting dead pixels and ϕ=0.87 and PPV=0.82 when detecting hot pixels.

### 5.3. Post-Layout Simulation Results

We randomly selected 500 images with different levels of brightness, contrast, and dynamic range from the Linnaeus 5 dataset [32] to create the images used to evaluate the detection and correction algorithms. This set of test images is different from the set of training images used to obtain the values of $P_{TL}$ and $P_{TH}$. All images have the same 128×128 resolution size, matching the proposed CMOS readout circuit array size. The most common type of defective pixel is the random distribution of dead and hot pixels in an image. In this paper, we use two variants of this type of defective pixel: 0.5% randomness (0.5% of all pixels in the image are defective), and 1.0% randomness (1.0% of all pixels in the image are defective). Furthermore, we impose the requirement that the dead pixel and hot pixel ranges be the lowest and highest 5%, respectively, of the full dynamic range. These values are consistent with the evaluation of other algorithms reported in the literature [12,13,17,18,19].

Figure 11 shows a post-layout simulation of the defective pixel detection circuit. The graphs in the figure show the voltages at four nodes of the circuit shown in Figure 3: the output $V_{C}$ of configurable CTIA, the reference inputs $V_{1}$ and $V_{2}$ of the comparators COMP1 and COMP2, and the output $V_{P}$, which is a logic 1 when the pixel is not defective. Figure 11a shows an example where the input pixel is not defective and passes the test in Line Four of Algorithm 1 ($P_{L}<P_{0}<P_{H}$). At t=0, FF1 is preset to 1 and capacitor $C_{int}$ is discharged. Between t=0 and t=12μs, the circuit operates in Phase 1, integrating the value of the input pixel $P_{0}$, and the switches at the input of COMP1 and COMP2 select $V_{1}=V_{H}$ and $V_{2}=V_{L}$ as reference voltages. At t=12μs, both comparators output a logic 1, because $P_{H}>P_{0}$ and $P_{0}>P_{L}$; thus, FF1 stores a logic 0 and $V_{P}=1$. At the same time, switch SWSH is opened to sample the pixel value. At t=12.25μs, the CTIA changes its integration direction and the circuit initiates Phase 2. Switch SWP0 opens and, for the next four 1μs-intervals, switches SWP2, SWP5, SWP7, and SWP4 select the neighboring pixels as inputs to the CTIA. At t=14μs, the input switches of COMP1 and COMP2 select $V_{1}=V_{TL}$ and $V_{2}=V_{TH}$. At t=16.25μs, Phase 2 ends and COMP1 and COMP2 output a logic 1 (the input pixel $P_{0}$ passes the test in Line Eight of Algorithm 1), and $V_{P}=1$.

Note that in Figure 11a it is actually irrelevant whether or not $P_{0}$ passes the test in Line Eight, because it has already passed the test in Line Four. In contrast, Figure 11b shows a case where the $P_{0}$ does not pass the test in Line Four of Algorithm 1, because $P_{0}>P_{H}$. Nevertheless, the pixel is not defective because it does pass the test in Line Eight. At the end of Phase 1, COMP1 outputs a logic 0 and COMP2 outputs a 1; therefore FF1 stores a logic 1 (marking the pixel as potentially defective) and $V_{P}=0$. However, at the end of Phase 2 both comparators output a logic 1, because the pixel passes the test in Line Eight; thus, the NAND1 in Figure 3 outputs a logic 0 and $V_{P}=1$, marking the pixel as not defective.

Figure 11c,d present examples where $P_{0}$ does not pass the test in Line Four of Algorithm 1, here because $P_{0}<P_{L}$ or $P_{0}>P_{H}$, respectively. Furthermore, in both examples $P_{0}$ does not pass the test in Line Eight, because $P_{0}-\frac{1}{3}P_{est}<P_{TL}$ or $P_{0}-\frac{1}{3}P_{est}>P_{TH}+\frac{2}{3}P_{max}$, respectively. In the case illustrated in Figure 11c, at the end of Phase 1 COMP1 outputs a logic 1 and COMP2 outputs a 0 ($P_{0}<P_{L}$). Therefore, FF1 stores a logic 1 and $V_{P}=0$. At the end of Phase 2, COMP1 outputs a 0 because $P_{0}-\frac{1}{3}P_{est}<P_{TL}$, while COMP2 outputs a 1. Thus, the first NAND in Figure 3 outputs a logic 1 and $V_{P}=0$, marking the pixel as defective. In the case shown in Figure 11d, at the end of Phase 1, COMP1 outputs a logic 0 ($P_{0}>P_{H}$) and COMP2 outputs a 1; thus, FF1 stores a logic 1 and $V_{P}=0$. At the end of Phase 2, COMP1 outputs a logic 1 and COMP2 outputs a 0, because $P_{0}-\frac{1}{3}P_{est}>P_{TH}+\frac{2}{3}P_{max}$. Thus, the first NAND in Figure 3 outputs a logic 1 and $V_{P}=0$, marking the pixel as defective.

Figure 12 shows a post-layout simulation of the median computation in the defective pixel correction circuit shown in Figure 5. In this test, we simulate the readout of five consecutive pixels of a column in the image. For each central pixel $P_{0}$, the row-select circuit outputs the voltage values of its four neighbors, denoted as V2, V5, V7, and V4 in the figure. For each of the neighbors, their voltage value is provided by the sample-and-hold circuit shown at the bottom right corner of Figure 3. As shown in Figure 12, we select a new pixel every 1μs; then, after a delay of 150ns, the output voltage $V_{out}$ settles at the median value shown in Equation (Equation 17), with a worst-case error of 14.8% in the pixel range. The delay is introduced by the minimum, maximum, and buffer circuits shown in Figure 5. The worst-case delay occurs when the minimum or maximum circuit at the input changes its output between two consecutive pixels (e.g., when first V2>V5 and next V5>V2) and that change propagates through to $V_{out}$. Moreover, the delay is maximal when the output voltage swing is close to $V_{dd}$. In this case the worst-case settling time for $V_{out}$ is 800 ns.

Figure 11 shows that the defective pixel detection circuit operates in 16.25μs, plus 50 ns to reset the integration capacitor after each detection. This circuit operates in parallel for all pixels in the image. During readout, the row-select circuit has a delay of 60 ns, the worst-case delay of the correction circuit is 800 ns at column level, and the column-select circuit has a delay of 80 ns. In this case, the readout time for the complete array is 1.44 ms, which allows for acquiring and processing images at a maximum frame rate of 694 fps.

### 5.4. Algorithm and Circuit Performance

Figure 13 illustrates the operation of the proposed algorithm and circuit. Figure 13a shows three images taken from the Linnaeus 5 dataset [32] which were contaminated with salt-and-pepper noise randomly affecting 1% of the pixels. Dead pixels were generated in a range of $\left[0,0.05P_{max}\right]$, while hot pixels were generated in a range of $\left[0.95P_{max},P_{max}\right]$. Figure 13b,c show the outputs of a software implementation of Algorithm 1 and our detection/correction circuit, respectively. In Figure 13b,c, false positive detections are highlighted in red and false negatives are marked in yellow.

To produce the images in Figure 13c, pixel values in the input image were converted to currents in the range from 0 to 14 nA in order to simulate the operation of a photodetector. Then, we ran a post-layout Spice simulation of the circuits described in Section 4 to produce the output pixel values and the list of pixels marked as defective by the circuit. Finally, we converted the output pixel voltages in the range from 0 to 3.3 V to digital values between 0 and 255 in order to display the output images.

A visual inspection of Figure 13 shows that the algorithm detects and corrects most defective pixels. Moreover, our pixel-correction circuit produces an output that is visually almost identical to the software implementation of the algorithm.

Figure 14 shows an enlarged version of the 35×85-pixel rectangle highlighted in cyan in the first image of Figure 13. Figure 14b,c show the output of the algorithm and correction circuit, respectively, with false positive detections highlighted in red and false negatives marked in green. Out of the 35 defective pixels in the image, the algorithm correctly detects 34 and the circuit detects 33. Moreover, out of the 2940 good pixels, the algorithm and circuit produce only seven false positives, which differ in only one detection. All the false positives and false negatives in Figure 14 correspond to dead pixels, which is explained by the mostly dark background in the image, and these do not significantly affect the appearance of the image.

Figure 15 compares the performance of our algorithm and hardware implementation to six other defective pixel detection algorithms, namely, those proposed by Chan [17], Yongji [19], Cho [18], the original algorithm proposed by Tchendjou et al., its simplification [12] (Tchendjou1 and Tchendjou2), and the median-based version of Tchendjou1 [13] (Tchendjou3), which does not requires dataset-dependent parameters. The figure plots the Se, Sp, PPV, and ϕ metrics, defined in Section 5.2, averaged over 500 128×128-pixel images randomly selected from the Linnaeus 5 dataset [32], to which we added salt-and-pepper noise affecting 0.5% and 1.0% of the pixels. These images are different from the 250 images used to determine the values of $P_{TL}$ and $P_{TH}$ in Section 5.2. The hardware implementation results were obtained from post-layout Spice simulations.

Figure 15a shows that Cho achieves the highest sensitivity (96.87%), though all the algorithms except for Yongji achieve sensitivity above 93%. In Figure 15b–d, it can be seen that our algorithm and its hardware implementation achieve the highest performance based on the Sp, PPV, and ϕ metrics. In summary, the classification performance of our algorithm is competitive with the state-of-the-art algorithms compared in Figure 15, while being based on simple arithmetical operations that can be efficiently implemented in analog hardware with comparable performance to software implementations.

To assess the combined performance of the detection and correction algorithms and their hardware implementations, we used the Peak Signal to Noise Ratio (PSNR) and Image Enhancement Factor (IEF) metrics. The PSNR metric compares a reference image $I_{0}$ without defective pixels to an image $I_{R}$ which results from applying a correction algorithm to a noisy version of $I_{0}$. The PSNR is defined by Equation (Equation 24):(24)$$PSNR=10\times log\left(\frac{P_{max}^{2}}{MSE}\right),$$
where $P_{max}$ is the maximum pixel value and MSE is the Mean Square Error, provided by Equation (Equation 25):(25)$$MSE=\frac{1}{M\times N}\sum_{i=1}^{M}\sum_{j=1}^{N}{\left(I_{0}\left(i,j\right)-I_{R}\left(i,j\right)\right)}^{2},$$
where M and N are the dimensions of the images.

IEF is computed as the ratio between the MSE of the noisy image and the MSE of the corrected image, using the noiseless image as a reference, as shown in Equation (Equation 26):(26)$$IEF=\frac{\sum_{i=1}^{M}\sum_{j=1}^{N}{\left(I_{N}\left(i,j\right)-I_{0}\left(i,j\right)\right)}^{2}}{\sum_{i=1}^{M}\sum_{j=1}^{N}{\left(I_{R}\left(i,j\right)-I_{0}\left(i,j\right)\right)}^{2}}.$$
where $I_{0}$ and $I_{R}$ are the noiseless and corrected images, respectively, and $I_{N}$ is the noisy image. If the correction algorithm reduces the MSE, then IEF>1; conversely, if the algorithm increases the MSE, then IEF<1.

Figure 16 compares the PSNR and IEF of our detection and correction algorithms and their hardware implementations to five algorithms proposed in the literature, labeled in the image as Chan [17], Yongji [19], Cho [18], Tchendjou1 [12], and Tchendjou3 [13]. In all cases, the algorithms performed defective pixel detection and correction on the same set of images used to generate the results shown in Figure 15. The hardware implementation results were obtained from post-layout Spice simulations of the circuits described in Section 4. Figure 16 shows that our algorithm performs better than all the other detection/correction methods as measured by both metrics. Our PSNR results are 11–15% higher than Tchendjou1 and Tchendjou3, and are more than 42% higher than the other three methods. In addition, the IEF achieved by our algorithm is more than 79% higher than the value achieved by Tchendjou1 and Tchendjou3, and more than 38 times the IEF achieved by Chan, Yongji, and Cho. These three methods produce a significant number of false positives, as shown in the PPV results in Figure 15c. The IEF metric is highly sensitive to the number of false positives, resulting in significant degradation in the performance of these three methods. Lastly, it can be observed that the PSNR of the analog hardware implementation is 1–3.7% worse than the algorithm, and the IEF is 12–17% lower. This is mainly due to charge injection in the detection circuit and the nonlinearities and offsets in the response of the detection and median computation circuits.

## 6. Conclusions

In this paper, we have presented an algorithm for defective pixel detection/correction along with its analog-CMOS hardware implementation. Our algorithm detects defective pixels using a simplification of the algorithm presented in [12] that replaces a pixel’s value with the median value of its four neighbors. The hardware implements defective pixel detection at the pixel level during photocurrent integration and corrects defective values at the column level during pixel readout.

An evaluation using 500 images with 0.5% and 1% random defective pixels shows that our algorithm outperforms similar methods published in the literature. Post-layout simulations of the hardware implementation of the algorithm achieves slightly lower performance than the software implementation, though it continues to perform better than the software versions of the other published methods. Similar to other detection/correction algorithms based on local statistics, our algorithm is sensitive to the occurrence of two or more defective pixels within a small window. In these cases, the estimations computed by the algorithms may be biased by neighboring defective pixels.

Performing arithmetical operations at the pixel level allows our circuit to detect incorrect values for all pixels in the imager in parallel. Moreover, using photocurrent integration to perform the operations reduces the impact on the die area compared to using dedicated arithmetic circuits. Although we use dedicated circuits to replace the defective pixel values, they are placed at each image column, reducing their impact on total circuit area while achieving column-level parallelism.

We are currently extending our work to other algorithms that can exploit pixel-level parallelism in hardware using the same approach presented in this paper. In particular, we are focusing on feature extraction using local gradients, local image filters, and non-uniformity correction for infrared image sensors.

## Figures and Tables

**Figure 1 sensors-23-00934-f001:**
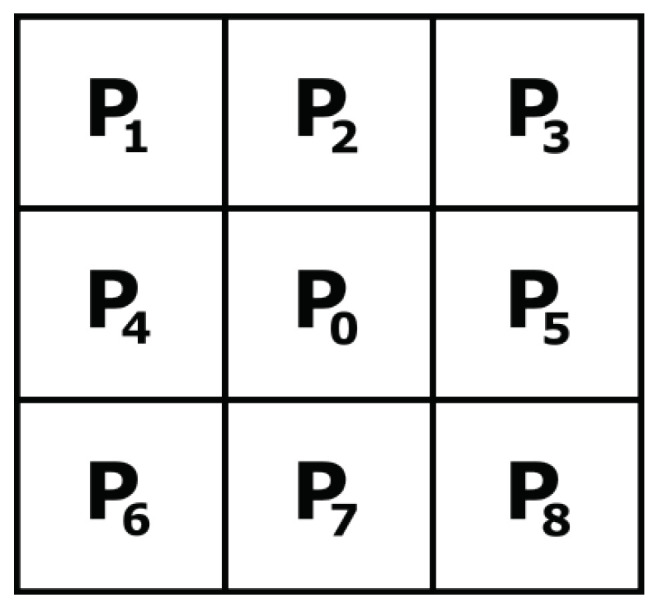
3×3-pixel window used by the defective pixel detection algorithm; $P_{0}$ is the pixel under test and $P_{1}$–$P_{8}$ are its neighbors.

**Figure 2 sensors-23-00934-f002:**
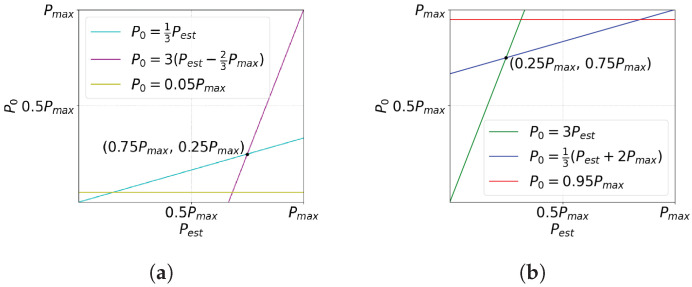
Detecting dead and hot pixels using Equations (Equation 6)–(Equation 9): (**a**) Dead pixels: Equations (Equation 6) and (Equation 7). When $P_{est}<0.05P_{max}$ and Equation (Equation 6) is true, Equation (Equation 7) is true. (**b**) Hot pixels: Equations (Equation 8) and (Equation 9). When $P_{est}>0.95P_{max}$ and Equation (Equation 9) is true, Equation (Equation 8) is true.

**Figure 3 sensors-23-00934-f003:**
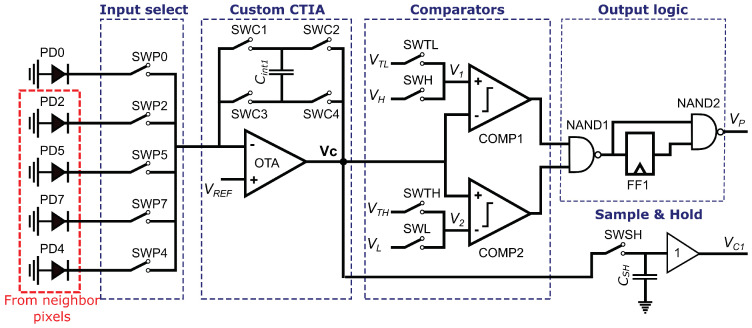
Defective pixel detection circuit.

**Figure 4 sensors-23-00934-f004:**
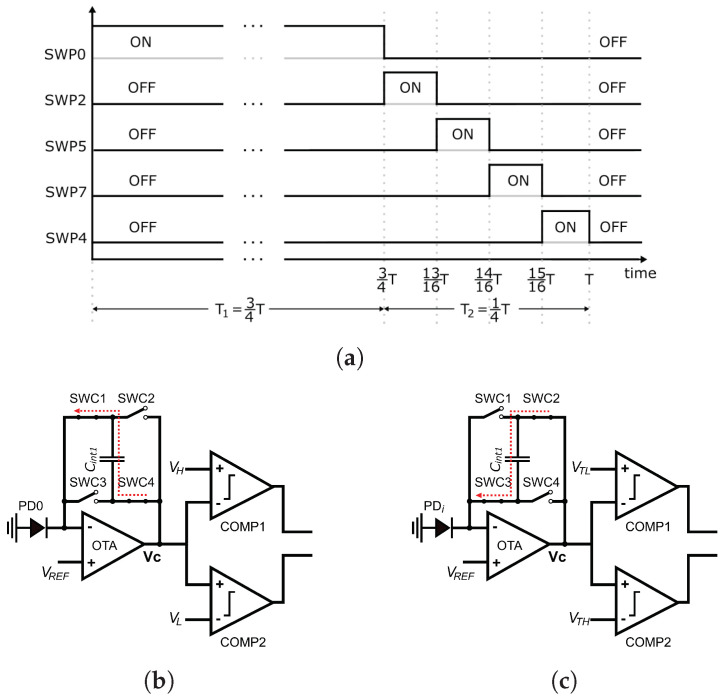
Input-select switch states and equivalent circuits for defective pixel detection during Phases 1 and 2 of circuit operation. The output logic stage is omitted for clarity. (**a**) Input-select switch states during Phases 1 and 2. (**b)** Equivalent circuit during Phase 1. (**c**) Equivalent circuit during Phase 2. In (**c**), $PD_{i}$ alternates between PD2, PD5, PD7, and PD4 during Phase 2.

**Figure 5 sensors-23-00934-f005:**
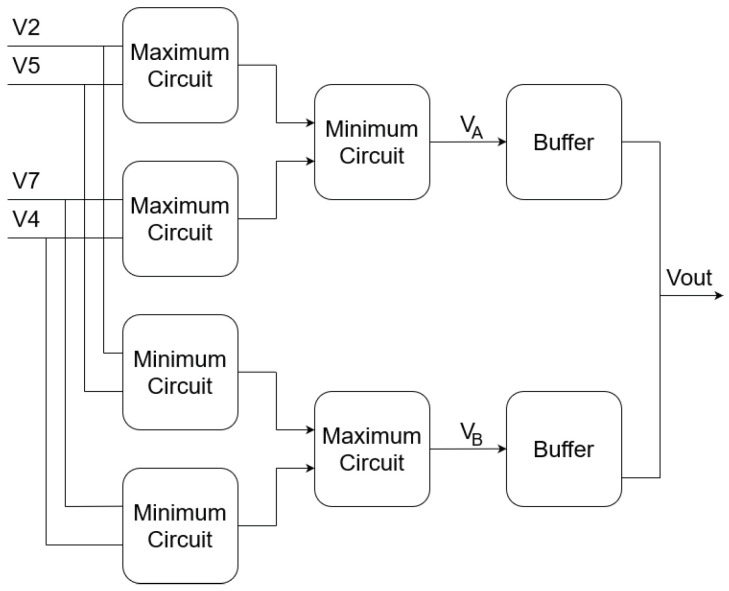
Block diagram of defective pixel correction circuit.

**Figure 6 sensors-23-00934-f006:**
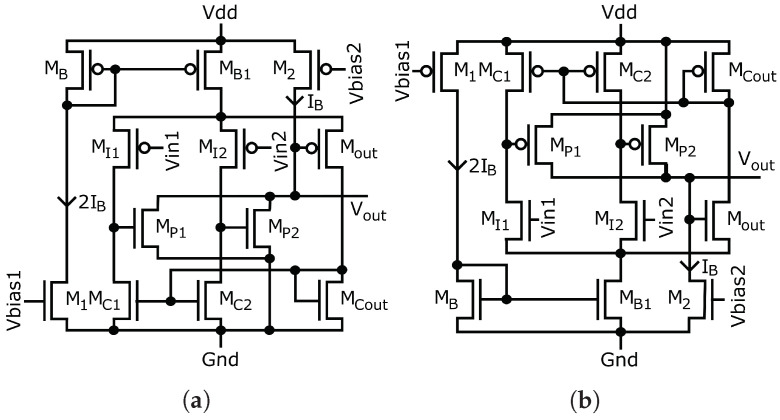
Circuit diagram to determine the minimum and maximum values between two voltages: (**a**) Minimum circuit diagram. (**b**) Maximum circuit diagram.

**Figure 7 sensors-23-00934-f007:**
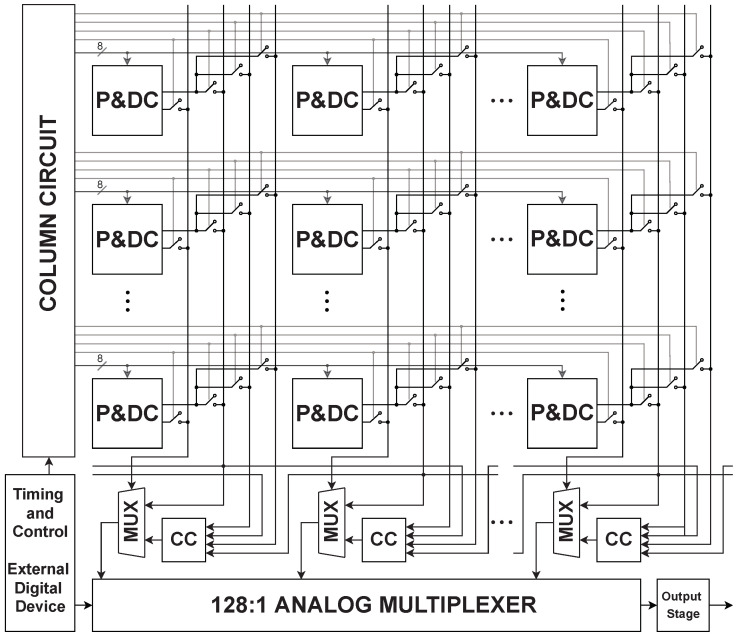
Smart pixel array.

**Figure 8 sensors-23-00934-f008:**
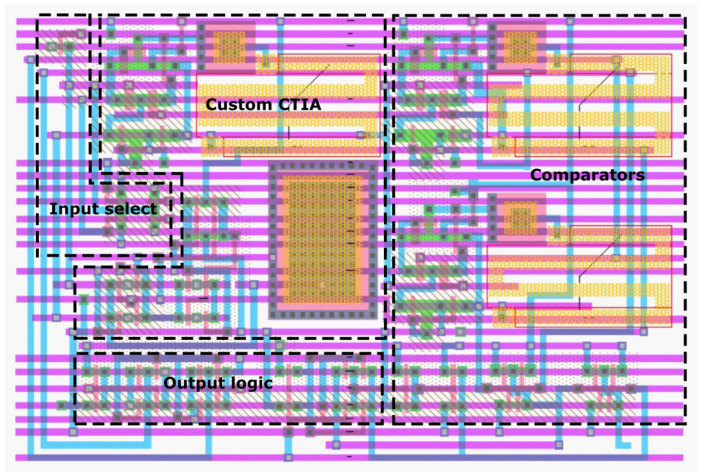
Layout of the defective pixel detection circuit.

**Figure 9 sensors-23-00934-f009:**
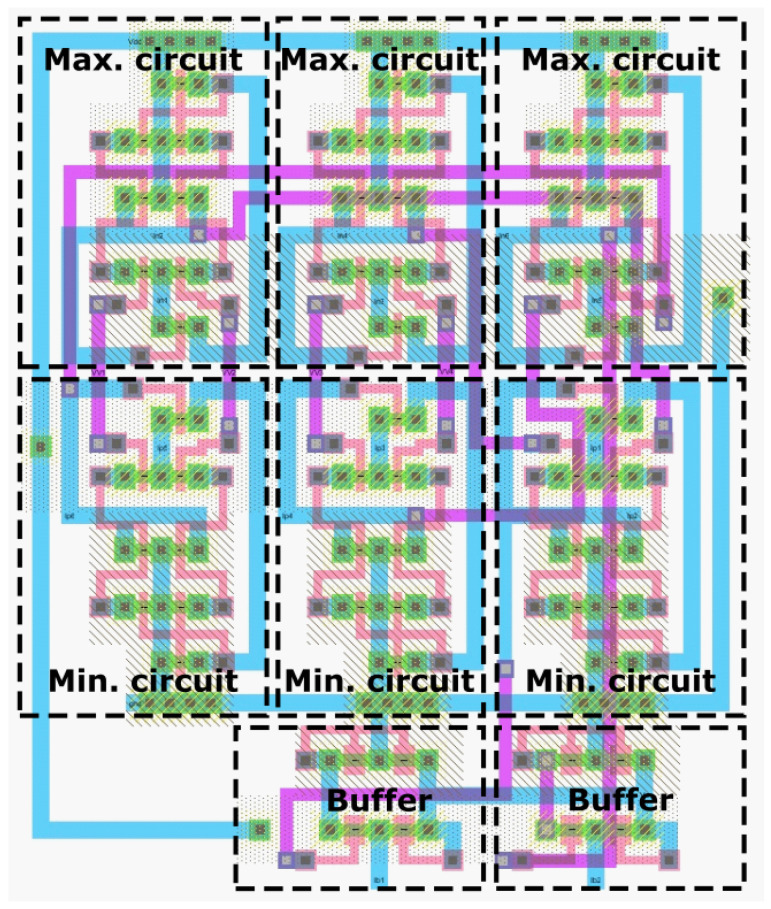
Layout of defective pixel correction circuit.

**Figure 10 sensors-23-00934-f010:**
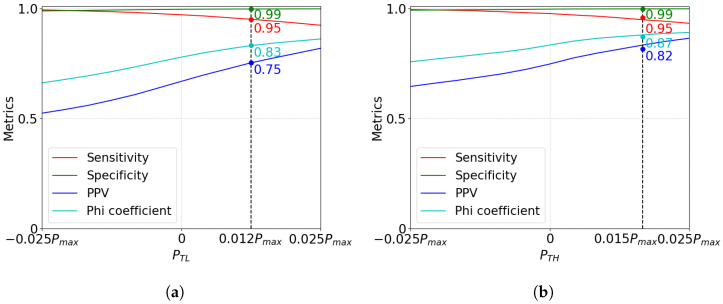
Performance of the algorithm using Sensitivity, Specificity, PPV, and Phi coefficient as functions of $P_{TL}$ and $P_{TH}$: (**a**) Performance of the algorithm as a function of $P_{TL}$. (**b**) Performance of the algorithm as a function of $P_{TH}$.

**Figure 11 sensors-23-00934-f011:**
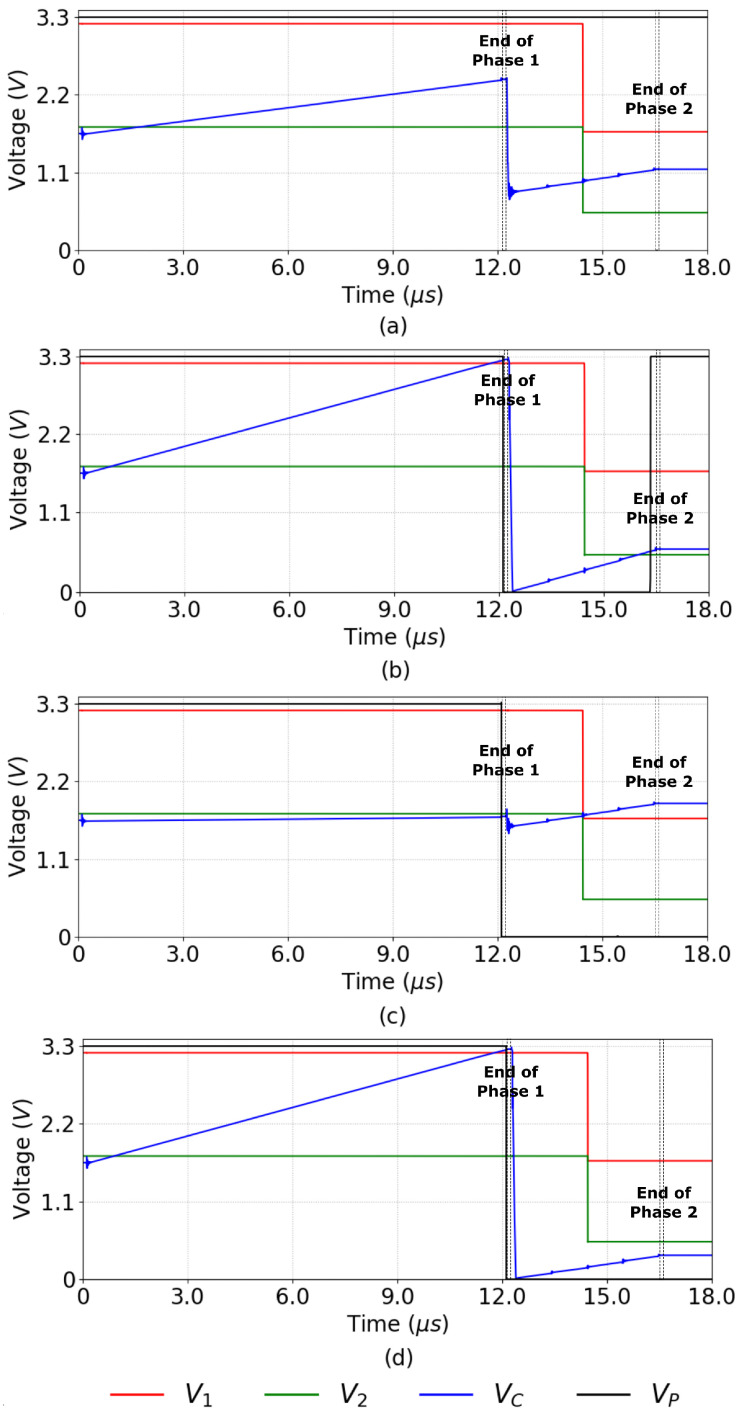
Post-layout simulation of defective pixel detection circuit: (**a**) a good pixel that passes the test in Line Four of Algorithm 1; (**b**) a good pixel that fails the test in Line Four of Algorithm 1; (**c**) a dead pixel. (**d**) a hot pixel.

**Figure 12 sensors-23-00934-f012:**
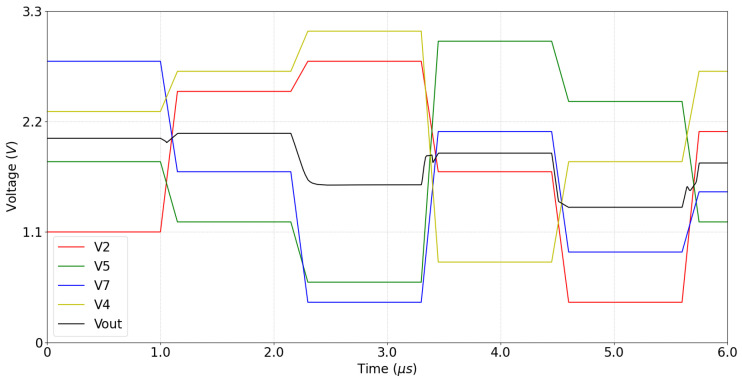
Post-layout simulation of the defective pixel correction circuit.

**Figure 13 sensors-23-00934-f013:**
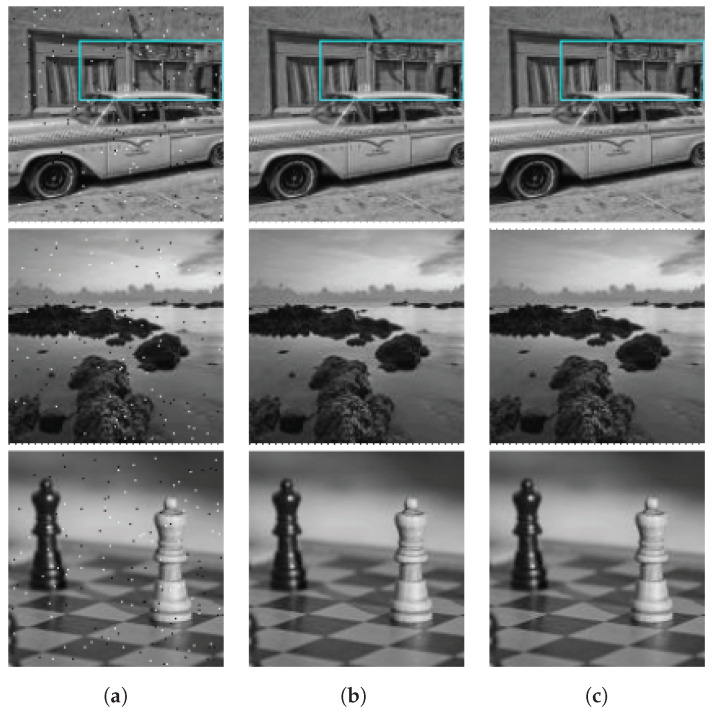
Original images taken from the Linnaeus 5 dataset [32] with 1.0% defective pixels added randomly along with example images showing the results of the defective pixel detection and correction process: (**a**) Original images with defective pixels. (**b**) Results of the proposed algorithm. (**c**) Results of the readout circuit.

**Figure 14 sensors-23-00934-f014:**
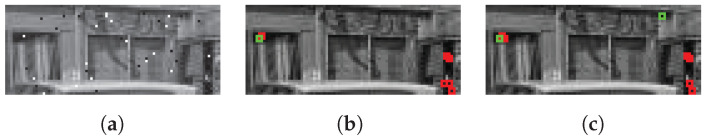
Zoom of a region of the original image taken from tthe Linnaeus 5 dataset [32] with false positive and false negative examples: (**a**) Noisy image. (**b**) Results of proposed algorithm. (**c**) Circuit output.

**Figure 15 sensors-23-00934-f015:**
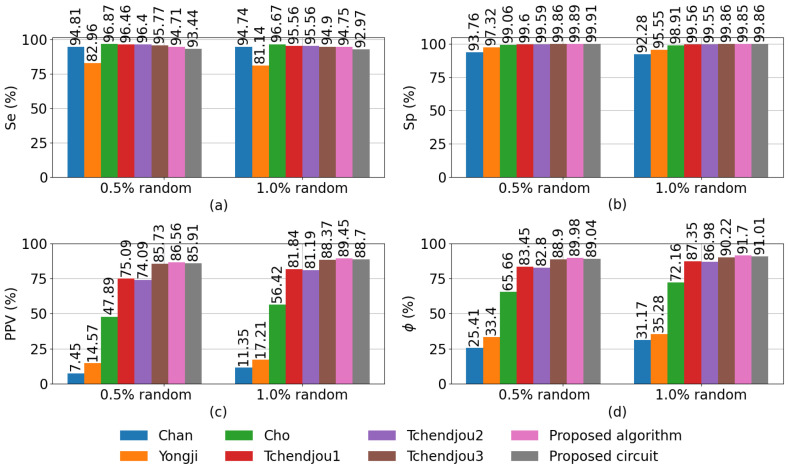
Comparison of defective pixel detection methods using (**a**) Sensitivity, (**b**) Specificity, (**c**) Positive Predictive Value (PPV), and (**d**) Phi coefficient.

**Figure 16 sensors-23-00934-f016:**
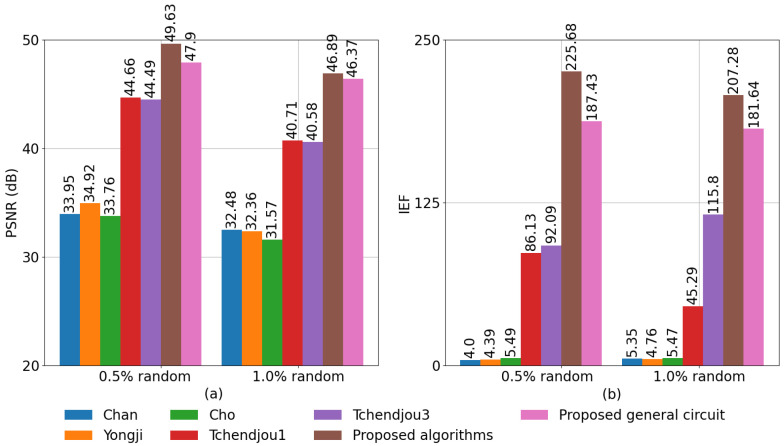
Comparison of defective pixel detection/correction methods using (**a**) PSNR and (**b**) IEF.

## Data Availability

This study uses the following publicly available dataset: Linnaeus 5 Dataset http://chaladze.com/l5/ (accessed on 28 April 2022).

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
