# Peer review of "A CMOS Image Readout Circuit with On-Chip Defective Pixel Detection and Correction"

_sensors, 2023, doi:10.3390/s23020934_

Round 1

Reviewer 1 Report

An on-chip defective pixel detection and correction method is proposed. The algorithm can be implemented by analog circuit. Better PSNR and IEF can be achieved. It is a very useful work for imaging. The main suggestions are listed as follows.

Detailed comments:

Abstract: L10. A 0.35um CMOS process is mentioned. However, 0.18um CMOS process may be used as presented in L357-365. Is it a writing error?

L15-16: It is suggested to simply present the related method for 294 fps.

Section3: The defective pixel is detected by comparing each pixel value to its four neighbors. It means each pixel should process the signal of its neighbors, so there will be many wires among pixels. How to place these wires in an efficient way? Are the crosstalk and difference among sensors are considered?

The expression in Figure 2 is different with equ.7. Is it a mistake?

Section 4.1, Figure 3. All neighbor pixels are integrated during phase 2. It means that these pixels will be shorted. Is it possible that the useful signal of the neighbor may be disrupted? The proper timing may be required.

L322. Why the positive-feedback loop is utilized? More expressions are expected. The circuits shown in Figure 6 are not very clear. Dots are hoped to be added to represent the connection. Mcout seams not be a part of the differential amplifier.

L348. Maybe levels should be layers.

The introduction of d is missed in the caption of figure 11.

L452. Which factor will influence the settling time of Vout? It can be seen that the voltage of V2, V4, V5 and V7 varies with time in Figure 12. Does it mean that there are current in the neighbor pixels, which is induced by light? The simulation setup is expected to be described.

Author Response

 Response to Reviewer 1 Comments

Below, we list our responses to each of the comments, and the changes we made to the manuscript to incorporate the suggestions of the reviewers. We believe the manuscript's quality has been improved by adding these changes and would like to thank the reviewers for their work. Additionally, we detected a few typos in the manuscript, which we have corrected. One of these typos was significant, as it listed the maximum frame rate achieved by our smart image sensor as 294 fps, when its correct value is 694 fps. We apologize for this error.

General comments

“An on-chip defective pixel detection and correction method is proposed. The algorithm can be implemented by analog circuit. Better PSNR and IEF can be achieved. It is a very useful work for imaging. The main suggestions are listed as follows.”

Point 1:

“Abstract: L10. A 0.35um CMOS process is mentioned. However, 0.18um CMOS process may be used as presented in L357-365. Is it a writing error?”

Response:

We thank the reviewer for noting a possible misleading sentence in the abstract.

All the results presented in the paper were obtained using technology files for a 0.35um, 2-poly, 3-metal process. However, we ported the layout of our circuits to a 0.18um process with metal-insulator-metal capacitors only to evaluate the scaleability of the design. Therefore, we computed only die area numbers for this process, and present them in Section 5. All other reported performance metrics of the CMOS implementation use the 0.35um process.

We modified the third paragraph of Section 5.1 to make our intention clearer: “Although the results presented in this section uses the 0.35um physical design described above, we also evaluated the scaleability of our circuits by porting the 0.35um physical design to a standard 0.18um CMOS process frequently used in the literature [62-65]. The 0.18um process uses a 1.8V supply voltage and features metal-insulator-metal capacitors with a capacitance per unit area of 2fF/um2. The total area of the defective pixel detection circuit in the 0.18um process is 30.1um by 45.2 um, which achieves a fill factor of 66.7% in the same 64um by 64um pixel. Without the comparators and output logic stages, the fill factor is 86.3%.”

Point 2:

“L15-16: It is suggested to simply present the related method for 294 fps.”

Response:

The results presented in the abstract and in Section 5 were indeed obtained at the maximum frame rate. To avoid confusion, we modified the fifth and sixth sentences of the abstract to clarify the maximum frame rate achieved by the design according to our post-layout simulations, and then report the defective-pixel detection/correction performance metrics, indicating that they were obtained with the circuit running at that frame rate.

We thank the reviewer for pointing out this possible confusion in the abstract.

Point 3:

Section3: The defective pixel is detected by comparing each pixel value to its four neighbors. It means each pixel should process the signal of its neighbors, so there will be many wires among pixels. How to place these wires in an efficient way? Are the crosstalk and difference among sensors are considered?

Response:

Although it is true that the detection algorithm uses the value of each pixel and its four neighbors, we implement eqns. 10-12 using time multiplexing during photocurrent integration. Using this scheme, the circuit integrates the current of only one pixel at each point in time. Therefore, only two wires are needed between each pair of neighboring pixels PA and PB: one to provide the current from PA to PB and one from PB to PA. Moreover, the two wires never transport current at the same time. This reduces crosstalk because when one wire is active, the other is in high impedance. Still, we use a grounded wire between each pair to provide shielding.

We added the following sentence to the second paragraph of Section 5.1: “It is worth noting that only two wires are required between neighboring cells, each transporting photocurrent in a different direction. This is because, as described in Section 4.1, the defective-pixel detection circuit implements Eqns.10-12 using time multiplexing, thus reading the photocurrent of only one pixel at a time. Moreover, because at most only one of the two wires operates in low impedance at the same time, the effects of crosstalk are diminished.

Point 4:

“The expression in Figure 2 is different with equ.7. Is it a mistake?”

Response:

We thank the reviewer for pointing out the error in Figure 2a. We changed the “+” sign in the equation shown in Figure 2a to “-”.

Point 5:

“Section 4.1, Figure 3. All neighbor pixels are integrated during phase 2. It means that these pixels will be shorted. Is it possible that the useful signal of the neighbor may be disrupted? The proper timing may be required.”

Response:

Although the value from four adjacent pixels is indeed integrated during phase 2 of the operation of the defective pixel detection circuit, the photocurrents from these four pixels are not shorted. Instead, the integration time during phase 2 is divided into 4 periods of equal duration and the currents are integrated in consecutive time periods using time-multiplexing. Thus, the integration process does not disrupt the signals from the neighboring pixels. Moreover, during phase 1, each pixel integrates its own local photocurrent and the resulting voltage is stored in a sample/hold circuit to provide the pixel output, which isolates it from the integration process in phase 2.

We thank the reviewer for pointing out that the explanation of the integration technique was not sufficiently clear. We added Fig. 4(a) to the paper to better illustrate the timing of the activation of switches SWP0, SWP2, SWP5, SWP7 and SWP4, which select the input to the CTIA of the detection circuit. We also updated the third and fourth paragraph of Section 4.1 to reference Fig. 4(a) during the description of the circuit and to clarify it operation.

Point 6:

“L322. Why the positive-feedback loop is utilized? More expressions are expected. The circuits shown in Figure 6 are not very clear. Dots are hoped to be added to represent the connection. Mcout seams not be a part of the differential amplifier.”

Response:

We thank the reviewer for this observation. We modified the third paragraph of Section 4.2 to provide a better explanation of the operation of the circuits and to make it clearer that we are using the circuits proposed by Soleimani [60]. We also modified the figure to clarify the connections between nodes. All connected nodes are marked with a dot, and when wires cross without a dot, they are not connected. We also made the lines thinner and the figure taller to make the diagram less cramped.

Point 7:

L348. Maybe levels should be layers.

Response:

The reviewer is correct. In the first paragraph in Section 5, we changed “levels” to “layers”.

Point 8:

“The introduction of d is missed in the caption of figure 11.”

Respnse:

We thank the reviewer for pointing out the missing information in the caption of Figure 11.

The caption now reads: “Post-layout simulation of defective pixel detection circuit. (a) Good pixel that passes the test in line 4 of Alg. 1. (b) Good pixel that fails the test in line 4 of Alg. 1. (c) Dead pixel. (d) Hot pixel.”

Point 9:

“L452. Which factor will influence the settling time of Vout? It can be seen that the voltage of V2, V4, V5 and V7 varies with time in Figure 12. Does it mean that there are current in the neighbor pixels, which is induced by light? The simulation setup is expected to be described.”

Response:

The settling time of the correction circuit output Vout is determined by the delay across the minimum, maximum and buffer circuits depicted in Fig. 5. The delay though each circuit depends on the total swing of its outputs when the circuit switches to the next row in the image, and the input voltages V2, V5, V7 and V4 change their value. For the values used in the experiment shown in Fig. 12, this delay is 150ns. However, the worst possible case occurs when the inputs to the “Maximum circuit” cell in Fig. 5 change so that its output also changes compared to the previous row (e.g. first V2 > V5 and next V5 > V2), and this change also switches the output of the "Minimum circuit” cell that follows it. The worst-case delay occurs when the output voltage swing is close to Vdd (3.3 V in our current design). In that case, the settling time of Vout reaches a maximum value of 800ns.

With respect to the second question, the changes in voltages V2, V5, V7 and V4 shown in Fig. 12 occur as the correction circuit (in each column) switches to the pixel in the next row. Fig. 12 shows the operation of one of the correction circuits during the readout of 5 consecutive pixels. While the circuit computes the corrected pixel value, the input voltages V2, V5, V7 and V4 do not vary because they are stored in the sample-and-hold circuit shown at the bottom right corner of Fig. 3 for the duration of the frame.

We modified the fifth and sixth paragraphs of Section 5.3 to reflect the clarifications made above.

Reviewer 2 Report

This paper proposes a defective pixel detection and correction algorithm and its implementation using CMOS analog circuits, which are integrated with the image sensor at the pixel and column levels. The proposal is interesting. The corrected results are also good.

There are some questions:

1. the corrected results are for images with 0.5% random defective pixels or 1.0% random defective pixels. If random defective pixels are more, even area pixels are defective, how is this proposal?

2. the proposal is for a slow-varying image. If the intensity of image is fast-varying, how is this proposal?

If the paper can explain these, that is better.

Author Response

Response to Reviewer 2 Comments

Below, we list our responses to each of the comments, and the changes we made to the manuscript to incorporate the suggestions of the reviewers. We believe the manuscript's quality has been improved by adding these changes and would like to thank the reviewers for their work. Additionally, we detected a few typos in the manuscript, which we have corrected. One of these typos was significant, as it listed the maximum frame rate achieved by our smart image sensor as 294 fps, when its correct value is 694 fps. We apologize for this error.

General comments

Point 1:

The corrected results are for images with 0.5% random defective pixels or 1.0% random defective pixels. If random defective pixels are more, even area pixels are defective, how is this proposal?

Response:

The reviewer is correct in pointing out that the performance of the algorithm depends at some point on the ratio of defective pixels. Indeed, a higher ratio of defective pixels increases the probability that one or more of the neighbors of a hot/dead pixel is also defective. This situation would decrease the performance of both our detection and correction algorithms, because both assume that the neighboring pixels are not defective. This effect is exacerbated when a block of pixels is defective. However, this is a common limitation of all algorithms that operate based on local statistics. In any case, our selection of 0.5% and 1% of defective pixels is consistent with other works reported in the literature.

We modified the paper to clarify the situation described above:

  • In the first paragraph of Section 5.3, we added the sentence “These values are consistent with the evaluation of other algorithms reported in the literature [12,13,17-19].”
  • In the second paragraph of the conclusions, we added “Like other detection/correction algorithms based on local statistics, our algorithm is sensitive to the occurrence of two or more defective pixels within a small window. In these cases, the estimations computed by the algorithms may be biased by neighboring defective pixels.

Point 2:

The proposal is for a slow-varying image. If the intensity of image is fast-varying, how is this proposal?

Response:

We thank the reviewer for this comment. Like the original algorithm proposed in [12], our algorithm operates within a single-image frame. Therefore, its performance does not depend on the speed at which the content of the video changes for each pixel. As discussed in section 5, the CMOS implementation shown in the paper operates at the maximum frame rate of 294 fps.

We modified the second paragraph of Section 3. Its second sentence now reads: “The original algorithm operates within a single-image frame, first defining a 3x3-pixel window centered on the pixel under test P0, as shown in Fig. 1.
